# Computing conditional entropies for quantum correlations

Peter Brown 1✉, Hamza Fawzi[2] & Omar Fawzi[1]

The rates of quantum cryptographic protocols are usually expressed in terms of a conditional entropy minimized over a certain set of quantum states. In particular, in the device-independent setting, the minimization is over all the quantum states jointly held by the adversary and the parties that are consistent with the statistics that are seen by the parties. Here, we introduce a method to approximate such entropic quantities. Applied to the setting of device-independent randomness generation and quantum key distribution, we obtain improvements on protocol rates in various settings. In particular, we find new upper bounds on the minimal global detection efficiency required to perform device-independent quantum key distribution without additional preprocessing. Furthermore, we show that our construction can be readily combined with the entropy accumulation theorem in order to establish full finite-key security proofs for these protocols.

[1] Univ Lyon, ENS Lyon, UCBL, CNRS, LIP, F-69342, Lyon Cedex 07, France. [2] DAMTP, University of Cambridge, Cambridge, UK. ✉email: peter.brown@ens-lyon.fr

Quantum cryptography is one of the most promising applications in the field of emerging quantum technologies having already seen commercial implementations. Using quantum systems it is possible to execute cryptographic protocols with security based on physical laws[1]—as opposed to assumptions of computational hardness. To date, much progress has been made in the development of new protocols and their respective security proofs. However, in real-world implementations such protocols are not infallible. Side-channel attacks arising from hardware imperfections or unreasonable assumptions in the security analysis can render the protocols useless[2]. While improvements in the hardware and more detailed security analyses can fix these issues, quantum theory also offers an alternative approach: device-independent (DI) cryptography.

Pioneered by the work of Mayers and Yao[3], device-independent cryptography circumvents the majority of side-channel attacks by offering security while making minimal assumptions about the hardware used in the protocol. Typically, we treat the devices used within an implementation of a DI protocol as black boxes. The remarkable fact that one can still securely perform certain cryptographic tasks on untrusted devices is a consequence of Bell-nonlocality[4]. In short, if an agent observes nonlocal correlations between two or more devices then they can infer restrictions on the systems used to produce them. It is then possible for the agent to infer additional desirable properties of their devices by analyzing this restricted class of systems. For example, it is known that all nonlocal correlations are necessarily random[5]. As a consequence, we can construct protocols for randomness generation (RNG)[6–8] and quantum key distribution (QKD)[3,9] with device-independent security.

A central problem in the development of new DI protocols is the question of how to calculate the rate of a protocol. That is, in DI-RNG how much randomness is generated or in DI-QKD how much secret key is generated per use of the device. For many DI protocols, including DI-RNG and DI-QKD, this problem reduces to minimizing the conditional von Neumann entropy over a set of quantum states that are characterized by restrictions on the correlations they can produce. Unfortunately, directly computing such an optimization is a highly non-trivial task. First, conditional entropies are non-linear functions of the states of a system and so the resulting optimization is in general non-convex and a naive optimization is not guaranteed to return a global optima. Moreover, as we are working device-independently we cannot assume any a priori bound on the dimensions of the systems used within the protocol. Nevertheless, in certain special cases the problem can be solved analytically[10]. However, the techniques used in the analysis of ref. [10] rely on particular algebraic properties of devices with binary inputs and binary outputs. As such, they do not generalize to more complex protocols with more inputs or outputs. This prompts the development of general numerical techniques to tackle this problem.

Simple numerical lower bounds on the von Neumann entropy minimization can be obtained through the min-entropy[11]. It was shown in refs. [12,13] that the analogous optimization of the min-entropy can be expressed as a noncommutative polynomial of measurement operators. This problem can then be relaxed to a semidefinite program (SDP) using the NPA hierarchy[14] which can then be solved efficiently. This approach gives a simple and efficient method to lower bound the rates of various DI tasks and has found widespread use in the analysis of DI protocols. Unfortunately, the min-entropy is in general much smaller than the von Neumann entropy and so this approach usually produces suboptimal results. More recently, the authors of ref. [15] extended the work of Coles et al.[16] to the device-independent setting. By viewing the objective function as an entropy gain between the systems producing the correlations they were able to construct a method to derive a noncommutative polynomial of the measurement operators that lower bounds the conditional von Neumann entropy. As in the case of the min-entropy approach, this can be approximated efficiently by an SDP. The numerical results presented in ref. [15] are very promising, providing significant improvements in the rates when compared to the min-entropy approach and also improving over the analytical results of ref. [10]. However, their approach is relatively computationally intensive requiring the optimization of a degree 6 polynomial in the simplest setting. For comparison, in protocols involving two devices, the min-entropy can always be computed using a polynomial of degree no larger than 2.

In this work we take a different approach, defining a new family of quantum Rényi divergences, the iterated mean (IM) divergences. The IM divergences are defined as solutions to certain SDPs and their constructions are inspired by the semidefinite representations of the weighted matrix geometric means[17]. These divergences define a corresponding family of conditional entropies that we show can be optimized device-independently using the NPA hierarchy and crucially they all form lower bounds on the conditional von Neumann entropy. We then apply these conditional entropies to the task of computing rates of DI randomness expansion (RE) and DI-QKD protocols. We compare the rates certified by our techniques with those certified by the min-entropy, the method of Tan et al.[15] and an analytical bound on $H(A|E)$ derived for the CHSH game[10]. Compared to the min-entropy bound, as will be shown in the examples we consider throughout the paper, our method almost always gives a significantly improved bound at a minor additional computational cost. Compared to the known analytical bound for CHSH, our method can be applied to a large family of protocols and this allows us to search for protocols that improve the various properties of interest. For example, by optimizing over a family of protocols with two inputs and two outputs per device we find a new upper bound on the minimal detection efficiency required to perform DI-QKD with a two-qubit system when we do not have an additional noisy preprocessing of the raw key[18,19] (see Fig. 1). Compared to the numerical work of Tan et al.[15], we find improvements in some regimes in the examples in which we could compare. However, it would appear that our method is more computationally efficient and hence can be applied to analyze a wider range of protocols. Finally, we demonstrate that our method can be used directly with the entropy accumulation theorem (EAT)[20,21] by constructing explicit min-tradeoff functions from the solutions of our optimizations. Applying the security proof blueprints developed in refs. [22,23] our techniques can be readily used together with the EAT to construct complete security proofs of many DI protocols. This property is again due to the simplicity of our method and it is unclear whether this can be done with the numerical method presented in ref. [15]. For all these reasons, we anticipate that the numerical tools developed here will lead to the development of better device-independent protocols.

## Results

In the following, we start by defining the IM divergences and their corresponding conditional entropies. We then show that the conditional entropies can be optimized in a device-independent manner and apply them to compute the rates of several device-independent tasks. In order to aid understanding, let us briefly introduce some notation. If $\mathcal{H}$ is a Hilbert space then $\mathscr{L}(\mathcal{H})$ denotes the set of linear operators from $\mathcal{H}$ to $\mathcal{H}$, $\mathscr{P}(\mathcal{H}) \subseteq \mathscr{L}(\mathcal{H})$ denotes the set of positive-semidefinite operators on $\mathcal{H}$ and

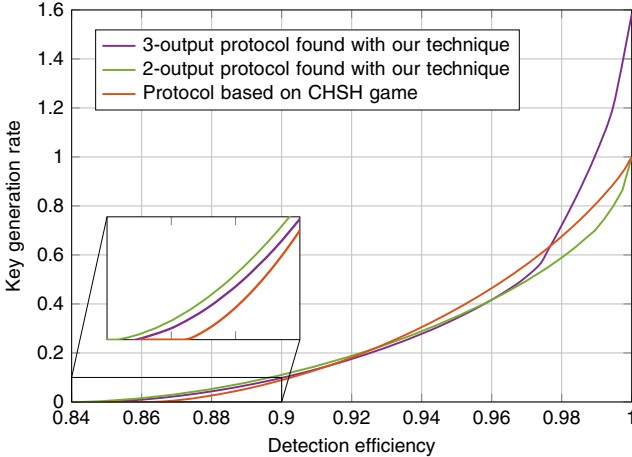

**Fig. 1 Comparing key rates of DI-QKD protocols without noisy preprocessing.** We compare the asymptotic key rates of a DI-QKD protocol based on the CHSH game, a DI-QKD protocol for 2-input 2-output devices and a DI-QKD protocol for 2-input 3-output devices when the respective devices used in the protocol are subject to inefficient detectors. Note that the key rates for the 3-output protocol can be smaller than the 2-output protocol in the regime of high noise as they were evaluated using different entropies from the iterated mean family. However, for low detection efficiency, we see that the 3-output protocol can achieve key rates of up to $\log(3)$ bits.

$\mathscr{D}(\mathcal{H}) \subseteq \mathscr{P}(\mathcal{H})$ denotes those with unit trace (quantum states). We write $\rho \ll \sigma$ to indicate that the support of $\rho$ is contained within the support of $\sigma$.

**Semidefinite programs for the iterated mean divergences.** The main technical contribution of this work is the introduction of a family of Rényi divergences that are amenable to device-independent optimization. Throughout the remainder of this work we define the sequence $\alpha_k := 1 + \frac{1}{2^k - 1}$ for $k \in \mathbb{N}$. We note that the name "iterated mean" comes from the expression that we establish later in Eq. (36).

**Definition 1** (Iterated mean divergences). Let $\mathcal{H}$ be a Hilbert space, $\rho \in \mathscr{D}(\mathcal{H})$, $\sigma \in \mathscr{P}(\mathcal{H})$ with $\rho \ll \sigma$ and let $\alpha_k = 1 + \frac{1}{2^k - 1}$ for each $k \in \mathbb{N}$. Then for each $k \geq 1$, we define the iterated mean divergence of order $\alpha_k$ as

$$D_{(\alpha_k)}(\rho \| \sigma) := \frac{1}{\alpha_k - 1} \log Q_{(\alpha_k)}(\rho \| \sigma),\qquad(1)$$

with $Q_{(\alpha_k)}(\rho \| \sigma)$ defined as

$$
\begin{aligned}
\max_{V_1,\dots,V_k,Z}\quad & \alpha_k \mathrm{Tr}\left[\rho \frac{(V_1 + V_1^*)}{2}\right] - (\alpha_k - 1)\mathrm{Tr}[\sigma Z]\\
\text{s.t}\quad & V_1 + V_1^* \geq 0\\
& \begin{pmatrix} I & V_i \\ V_i^* & \frac{(V_{i+1}+V_{i+1}^*)}{2} \end{pmatrix} \geq 0\\
& \begin{pmatrix} I & V_k \\ V_k^* & Z \end{pmatrix} \geq 0
\end{aligned}
\qquad(2)
$$

where the penultimate constraint is for all $1 \leq i \leq k - 1$ and the optimization varies over $V_1, \dots, V_k \in \mathscr{L}(\mathcal{H})$ and $Z \in \mathscr{P}(\mathcal{H})$. We may assume further that $Z \ll \sigma$ and $V_i \ll \sigma$ for each $i \in \{1, 2, \dots, k\}$. Note that we may equivalently write the

constraints as

$$V_1 + V_1^* \geq 0, \qquad \frac{V_2 + V_2^*}{2} \geq V_1^* V_1, \qquad \cdots \qquad Z \geq V_k^* V_k.$$

*Remark* 1 (Important property for device-independent optimization). The crucial property that makes these divergences well-adapted for device-independent optimization is the fact that $Q_{(\alpha_k)}(\rho \| \sigma)$ has a free variational formula as a supremum of linear functions in $\rho$ and $\sigma$. We say that $Q$ has a free variational formula if there exists $m, n \in \mathbb{N}$ and noncommutative Hermitian polynomials $p_1, \dots, p_n$ in the variables $(V_1, \dots, V_m)$ such that for any dimension $d \geq 1$, $\rho \in \mathscr{D}(\mathbb{C}^d)$ and $\sigma \in \mathscr{P}(\mathbb{C}^d)$

$$Q(\rho \| \sigma) = \max_{(V_1, V_2, \dots, V_m) \in S(d)} \mathrm{Tr}[V_1 \rho] + \mathrm{Tr}[V_2 \sigma],\qquad(3)$$

where the family of sets $\{S(d)\}_{d \in \mathbb{N}}$ are all defined using the same polynomials $p_1, \dots, p_n$, i.e.,

$$S(d) = \{(V_1, \dots, V_m) \in (\mathbb{C}^{d \times d})^m : p_j(V_1, \dots, V_m) \geq 0 \; \forall j \in \{1, \dots, n\}\}.\qquad(4)$$

We repeat that the important property is that the sets $S(d)$ describing the linear functions have a uniform description that is independent of the dimension $d$ (the polynomials $p_j$ are the same for all dimensions $d$). Such families of sets are studied in the area of free semialgebraic geometry (see e.g., refs. [24,25]). Note that the measured Rényi divergences have such a formulation as expressed in Eq. (29) (for rational values of $\alpha$), but these divergences can be smaller than the Umegaki divergence and thus cannot be used to give lower bounds on the von Neumann entropy. It remains an important open problem whether the sandwiched or the Petz divergences can be expressed using free variational formulas of the form (3). Here, we have introduced new divergences $D_{(\alpha_k)}$ that have this property by construction. Note that a representation as in Eq. (3) immediately establishes joint convexity of $Q$ (regardless of the freeness of the representation). As such finding a free variational formula for the sandwiched or the Petz divergences would provide a "dimension-free" proof of joint convexity and as these families are known to converge to $D$ as $\alpha \to 1$, such free variational formulas would lead to converging approximations for the von Neumann entropy that we aim to approximate. With the divergences $D_{(\alpha_k)}$, we can only guarantee convergence as $k \to \infty$ to the von Neumann entropy in the commuting case. In the general case, it remains open to determine the limit as $k \to \infty$ of $D_{(\alpha_k)}$.

Given a bipartite quantum state $\rho \in \mathscr{D}(AB)$ and a divergence $\mathbb{D}$ we may define a corresponding conditional entropy as $\mathbb{H}^{\downarrow}(A|B) = -\mathbb{D}(\rho_{AB} \| I_A \otimes \rho_B)$ and its optimized version as $\mathbb{H}^{\uparrow}(A|B) = \sup_{\sigma \in \mathscr{D}(B)} -\mathbb{D}(\rho_{AB} \| I_A \otimes \sigma_B)$. The following proposition gives an explicit characterization of $\mathbb{H}^{\uparrow}$ for the iterated mean divergences. We defer the proof of this proposition to the "Methods" section.

**Proposition 1.** *Let $\rho \in \mathscr{D}(AB)$. Then*

$$H^{\uparrow}_{(\alpha_k)}(A|B)_{\rho} = \frac{1}{1 - \alpha_k} \log Q^{\uparrow}_{(\alpha_k)}(\rho)\qquad(5)$$

with $Q^{\uparrow}_{(\alpha_k)}(\rho)$ defined as

$$
\begin{aligned}
\max_{V_1, \dots, V_k}\quad & \left(\mathrm{Tr}\left[\rho \frac{(V_1 + V_1^*)}{2}\right]\right)^{\alpha_k}\\
\text{s.t}\quad & \mathrm{Tr}_A[V_k^* V_k] \leq I_B\\
& V_1 + V_1^* \geq 0\\
& \begin{pmatrix} I & V_i \\ V_i^* & \frac{(V_{i+1}+V_{i+1}^*)}{2} \end{pmatrix} \geq 0,
\end{aligned}
\qquad(6)
$$

where the final constraint is for all $1 \leq i \leq k - 1$.

It can be shown (see the "Methods" section) that for each $\alpha_k$ and any pair $(\rho, \sigma)$, $D_{(\alpha_k)}(\rho \| \sigma) \geq \widetilde{D}_{\alpha_k}(\rho \| \sigma)$ where $\widetilde{D}_\alpha$ denotes the sandwiched Rényi divergence of order $\alpha$[26,27]. In turn, we have that for any bipartite state $\rho \in \mathscr{D}(AB)$, $H^\uparrow_{(\alpha_k)}(A|B)_\rho \leq \widetilde{H}^\uparrow_{\alpha_k}(A|B)_\rho$. Thus as $\widetilde{H}^\uparrow_\alpha(A|B) \leq H(A|B)$ for all $\alpha > 1$ we also have that the IM conditional entropies lower bound the conditional von Neumann entropy. Therefore we may use them to compute lower bounds on the rates of various device-independent protocols.

**Application to device-independent cryptography.** In the following, we consider the setup wherein there are two devices (which we refer to as Alice and Bob) that receive inputs $X$ and $Y$ from some finite alphabets $\mathcal{X}$ and $\mathcal{Y}$ and produce outputs $A$ and $B$ in some finite alphabets $\mathcal{A}$ and $\mathcal{B}$, respectively. We restrict to a bipartite setting for simplicity but our techniques can be readily extended to multipartite settings. During a single interaction, we assume that the devices operate in the following way. A bipartite quantum state $\rho_{Q_A Q_B} \in \mathscr{D}(Q_A Q_B)$ is shared between the two devices and in response to the inputs $x \in \mathcal{X}, y \in \mathcal{Y}$ the devices perform the POVMs $\{M_{a|x}\}_{a \in \mathcal{A}}$, $\{N_{b|y}\}_{b \in \mathcal{B}}$ on their respective systems. Inputs are chosen according to some fixed distribution $\mu : \mathcal{X} \times \mathcal{Y} \to [0, 1]$ that is known to all parties. The conditional probability distribution that describes the input–output behavior of the two devices is then given by

$$p(a, b|x, y) = \mathrm{Tr}\left[\rho_{Q_A Q_B}(M_{a|x} \otimes N_{b|y})\right]. \tag{7}$$

In addition, we allow for the presence of an adversarial party (Eve) who holds a purification of the quantum state initially shared between Alice and Bob, i.e., there is some pure quantum state $|\psi\rangle\langle\psi| \in \mathscr{D}(Q_A Q_B E)$ such that $\mathrm{Tr}_E[|\psi\rangle\langle\psi|] = \rho_{Q_A Q_B}$. Formally, this setting may be characterized by a tuple $(Q_A, Q_B, E, |\psi\rangle, \{M_{a|x}\}, \{N_{b|y}\})$ which we shall refer to as a strategy.

Let $\mathcal{C}$ be another finite alphabet and let $C : \mathcal{A} \times \mathcal{B} \times \mathcal{X} \times \mathcal{Y} \to \mathcal{C}$ be some function—this function will act as a statistical test on the devices. Given a probability distribution $q : \mathcal{C} \to [0, 1]$ we say that a conditional distribution $p_{AB|XY}$ is compatible with $q$ if for all $c \in \mathcal{C}$ we have

$$\sum_{abxy:C(a,b,x,y)=c} \mu(x, y) p(a, b|x, y) = q(c). \tag{8}$$

More generally we say that a strategy $S$ is compatible with the statistics $q$ if the conditional distribution induced by the strategy (cf. Eq. (7)) is compatible with $q$. For a given statistical test $C$ we denote the collection of all strategies that are compatible with the statistics $q$ by $\Sigma_C(q)$. The post-measurement state of a strategy $S = (Q_A, Q_B, E, |\psi\rangle, \{M_{a|x}\}, \{N_{b|y}\})$ is

$$\rho_{ABXYE} = \sum_{abxy} \mu(x, y) |abxy\rangle\langle abxy| \otimes \rho_E^{abxy} \tag{9}$$

where

$$\rho_E^{abxy} = \mathrm{Tr}_{Q_A Q_B}\left[(M_{a|x} \otimes N_{b|y} \otimes I_E)|\psi\rangle\langle\psi|\right]. \tag{10}$$

Let $\mathcal{P}(\mathcal{C})$ denote the set of all probability distributions on the alphabet $\mathcal{C}$. A global tradeoff function for the statistical test $C$ is a function $f : \mathcal{P}(\mathcal{C}) \to \mathbb{R}$ such that

$$f(q) \leq \inf_{\Sigma_C(q)} H(AB|XYE), \tag{11}$$

where the infimum is taken over post-measurement states of all strategies that are compatible with the statistics $q$. Similarly, we

say a function $f : \mathcal{P}(\mathcal{C}) \to \mathbb{R}$ is a local tradeoff function for the statistical test $C$ if it satisfies

$$f(q) \leq \inf_{\Sigma_C(q)} H(A|XE). \tag{12}$$

We shall now demonstrate how to compute device-independent lower bounds on Eqs. (11) and (12) using the conditional entropies $H^\uparrow_{(\alpha_k)}(AB|XYE)$. Furthermore, by replicating the tradeoff function constructions presented in ref. [28] for the min-entropy, we can also derive explicit affine tradeoff functions from the results of our optimizations. Therefore, the present analysis can be readily extended to a full security proof of a device-independent protocol through an application of the entropy accumulation theorem[20,21].

Following the device-independent setup described above, we look to evaluate the conditional entropies for classical-quantum states that arise from a measurement on some subsystem. Considering this scenario, the following lemma explains how we can rewrite $H^\uparrow_{(\alpha_k)}(AB|XYE)$ into a form which can then be relaxed to a semidefinite program via the NPA hierarchy and hence optimized in a device-independent manner. The basic idea is that if we have a cq-state $\rho_{AE}$ then it is sufficient to consider variables that are block diagonal, i.e., $V_i = \sum_a |a\rangle\langle a| \otimes V_{i,a}$. Furthermore, we can rewrite the objective function such that it explicitly depends on the POVM and the state $|\psi\rangle\langle\psi|$ used to generate the classical register. We defer the proof of this lemma to the "Methods" section.

**Lemma 1.** *Let $|\psi\rangle\langle\psi| \in \mathscr{D}(Q_A E)$, $\{M_a\}_{a \in \mathcal{A}}$ be a POVM on $Q_A$ and $\rho_{AE} = \sum_a |a\rangle\langle a| \otimes \rho_E(a)$ be a cq-state where $\rho_E(a) = \mathrm{Tr}_{Q_A}[(M_a \otimes I)|\psi\rangle\langle\psi|]$. Then, for each $k \in \mathbb{N}$ we have*

$$H^\uparrow_{(\alpha_k)}(A|E) = \frac{\alpha_k}{1 - \alpha_k} \log Q^{\mathrm{DI}}_{(\alpha_k)} \tag{13}$$

*with $Q^{\mathrm{DI}}_{(\alpha_k)}$ defined as*

$$\begin{aligned}
\max_{V_{i,a}: 1 \leq i \leq k, a, \in A_k} \quad & \sum_a \mathrm{Tr}\left[\left(M_a \otimes \frac{(V_{1,a} + V_{1,a}^*)}{2}\right)|\psi\rangle\langle\psi|\right] \\
\text{s.t} \quad & \sum_a V_{k,a}^* V_{k,a} \leq I_E \\
& V_{1,a} + V_{1,a}^* \geq 0 \\
& 2V_{i,a}^* V_{i,a} \leq V_{i+1,a} + V_{i+1,a}^*
\end{aligned} \tag{14}$$

*where the final two sets of constraints are for all $a \in \mathcal{A}$ and $1 \leq i \leq k - 1$.*

*Example* 1. For the post-measurement state of a strategy $S = (Q_A, Q_B, E, |\psi\rangle, \{M_{a|x}\}, \{N_{b|y}\})$ we have that $2^{-\frac{1}{2}H^\uparrow_{(2)}(AB|X=x,Y=y,E)}$ is equal to

$$\begin{aligned}
\max_{V_{a,b}: a \in A, b \in B} \quad & \sum_{ab} \mathrm{Tr}\left[\left(M_{a|x} \otimes N_{b|y} \otimes \frac{(V_{a,b} + V_{a,b}^*)}{2}\right)|\psi\rangle\langle\psi|\right] \\
\text{s.t} \quad & \sum_a V_{a,b}^* V_{a,b} \leq I_E \\
& V_{a,b} + V_{a,b}^* \geq 0
\end{aligned}. \tag{15}$$

Comparing this with the analogous optimization for the conditional min-entropy, $2^{-H_{\min}(AB|X=x,Y=y,E)}$,

$$\begin{aligned}
\max_{W_{a,b}: a \in A, b \in B} \quad & \sum_{ab} \mathrm{Tr}\left[\left(M_{a|x} \otimes N_{b|y} \otimes W_{ab}\right)|\psi\rangle\langle\psi|\right] \\
\text{s.t} \quad & \sum_a W_{ab} \leq I_E \\
& W_{ab} \geq 0
\end{aligned} \tag{16}$$

we see several similarities in the structure of the optimization.

The rewriting of $H^\uparrow_{(\alpha_k)}(A|E)$ in Lemma 1 still refers to an explicit pair of Hilbert spaces $Q_A$, $E$, and an explicit state $|\psi\rangle \in Q_A E$. In order to compute device-independent lower

bounds on the various entropic quantities we also take the supremum in Eq. (14) over all pairs of Hilbert spaces, and all operators and states on those Hilbert spaces. As mentioned previously, in order to approximate this extended optimization in an efficient manner it is possible to relax the optimization problem to a semidefinite program using the NPA hierarchy[14]. Indeed, we can optimize over moment matrices generated by the monomials $\{I\} \cup \{M_a\}_{a \in \mathcal{A}} \cup \{V_{i,a}, V_{i,a}^*\}_{1 \leq i \leq k, a \in \mathcal{A}}$. The operator inequalities can be replaced by localizing moment matrices and we can enforce that all $[M_a, V_{i,a'}] = 0$ for all $a, a' \in \mathcal{A}$ and $1 \leq i \leq k$. We are also free to impose statistical constraints on our devices, e.g., a Bell-inequality violation. When calculating $H_{(2)}^{\uparrow}$ there are also additional constraints that we may add in certain cases to help speed up convergence. These are analogous to assuming that the POVM operators are projective when computing $H_{\min}$ (see Remark 4 for further details).

A more detailed explanation of the SDP implementation is given in the Supplementary Information. To help facilitate the use of our techniques we also provide a few coded examples[29]. The NPA hierarchy relaxations were computed using the python package NCPOL2SDPA[30] and all SDPs were solved using the Mosek solver[31]. For simplicity we shall only consider the entropy of some fixed inputs $(X, Y) = (x_0, y_0)$—this reflects the scenario usually considered in device-independent protocols where certain inputs are dedicated to generating secret key or randomness. For this reason, in the following application subsections, we will abuse notation and for a conditional entropy $\mathbb{H}$, we will write $\mathbb{H}(AB|E)$ and $\mathbb{H}(A|E)$ instead of $\mathbb{H}(AB|X = x_0, Y = y_0, E)$ and $\mathbb{H}(A|X = x_0, E)$, respectively, where the choice of $x_0, y_0$ will be clear from the context or otherwise explicitly stated. To distinguish devices with different numbers of inputs and outputs we use the shorthand *abcd*-scenario to denote the situation wherein Alice's device has *a* inputs and *c* outputs and Bob's device has *b* inputs and *d* outputs.

**Application: Randomness certification.** We applied the semi-definite relaxations of $Q_{(\alpha_k)}^{DI}$ to compute device-independent lower bounds on $H_{(4/3)}^{\uparrow}(AB|E)$ and $H_{(2)}^{\uparrow}(AB|E)$ for different statistical constraints. First, we considered the CHSH game which is defined by the function

$$C_{CHSH}(a, b, x, y) = \begin{cases} 1 & \text{if } a \oplus b = xy \\ 0 & \text{otherwise.} \end{cases} \tag{17}$$

In addition to this we also considered the situation where the devices are constrained by their full conditional distribution, i.e., we record each input–output tuple as a separate score $C: (a, b, x, y) \mapsto (a, b, x, y)$. We compared these to a tight analytical bound on the local von Neumann entropy $H(A|X = 0, E)$ which is known for the CHSH game[10,22], numerical lower bounds on $H_{\min}$ and the recent numerical lower bounds on the von Neumann entropy which were developed in ref. [15] (we refer to these latter bounds as the TSGPL bounds). For both devices constrained by the CHSH game and devices constrained by their full conditional distribution we evaluate the entropy for the inputs $(x_0, y_0) = (0, 0)$.

In Fig. 2, we plot lower bounds on the global entropies of Alice and Bob when their devices are constrained to achieve a minimal CHSH score. In the plot we observe a separation between the three curves that we compute numerically. That is, as we decrease $\alpha_k$ toward 1 we see visible improvements on the certifiable rates. For larger CHSH scores we observe that the lower bounds for both $H_{(4/3)}^{\uparrow}(AB|E)$ and $H_{(2)}^{\uparrow}(AB|E)$ can be used to certify

substantially more randomness than $H_{\min}(AB|E)$. However, all three curves eventually drop below the randomness certified by the tight analytical bound on $H(A|E)$.

Recent device-independent experiments[32,33] have relied on measuring entangled photons in order to generate nonlocal correlations. A major source of noise in these systems comes from inefficient detectors or losses during transmission of the photons. We model this noise by a single parameter $\eta \in [0, 1]$ which characterizes the probability that after a photon has been produced by the source it is successfully transmitted and detected. For simplicity, we use the same $\eta$ for the photons of each party. In order to avoid a detection loophole in the experiment[34], all failed detection events are recorded as the outcome 0. This noise transforms the noiseless conditional probability distribution produced by the two parties in the following way

$$\begin{aligned} p(a, b|x, y) \mapsto \eta^2 p(a, b|x, y) + \eta(1 - \eta)(\delta_{a0} p(b|y) \\ + \delta_{b0} p(a|x)) + \delta_{a0} \delta_{b0} (1 - \eta)^2, \end{aligned} \tag{18}$$

where $\delta_{ij}$ is the Kronecker delta function. In order to generate valid quantum probability distributions we consider a two-qubit setup with a state $|\psi_\theta\rangle = \cos(\theta)|00\rangle + \sin(\theta)|11\rangle$ with $\theta \in (0, \pi/4)$ and two-outcome qubit POVMs of the form $\{M, I - M\}$ where $M = |v\rangle\langle v|$ with $|v\rangle = \cos(\phi/2)|0\rangle + \sin(\phi/2)|1\rangle$ and $\phi \in (-\pi, \pi]$. We assume that $\mathcal{A} = \mathcal{B} = \mathcal{X} = \mathcal{Y} = \{0, 1\}$.

In Fig. 3, we compare lower bounds on the randomness certified by the different conditional entropies when the devices operate with inefficient detectors. We see that as before $H_{(4/3)}^{\uparrow}(AB|E)$ and $H_{(2)}^{\uparrow}(A|E)$ can be much larger than $H_{\min}(AB|E)$ and that the difference is more pronounced in this case. Moreover, by constraining the devices by the full conditional distribution we find a much larger improvement over the analytical bound on $H(A|E)$ which is only constrained by the CHSH game. Through our optimization over two-qubit systems we were also able to find systems that can certify the upper bound of two bits of randomness in the noiseless case. Unlike in Fig. 2 we find in this case a negligible difference between the randomness certified by $H_{(4/3)}^{\uparrow}(AB|E)$ and $H_{(2)}^{\uparrow}(AB|E)$. Comparing with the TSGPL bound we find that our optimized curves can certify more randomness in the lower noise regimes ($\eta > 0.92$). However for higher noise the TSGPL bound outperforms our method in this setting.

**Application: Quantum key distribution.** Continuing the comparison of entropy bounds for systems with inefficient detectors, we look at how this noise affects the rates of DI-QKD. Again we will consider devices that are constrained by the full conditional probability distribution, as was the case in Fig. 3. However, here we consider two separate setups. First, we look at the 2322-scenario, i.e., $\mathcal{A} = \mathcal{B} = \mathcal{X} = \{0, 1\}$ and $\mathcal{Y} = \{0, 1, 2\}$. We give Bob a third input which will act as his key-generation input, e.g., the key will be generated from the outputs of the devices on the input pair $(X, Y) = (0, 2)$. Ideally, the correlations between Alice and Bob on this input pair are such that $H(A|B)$ is small. We generate the correlations of these devices with the same two-qubit model introduced previously. As a novel comparison, we also look at the 2333-scenario, i.e., $\mathcal{A} = \mathcal{B} = \{0, 1, 2\}$, $\mathcal{X} = \{0, 1\}$, and $\mathcal{Y} = \{0, 1, 2\}$. We generate probability distributions for these devices using a two-qutrit model. As before, we assume an explicit model for the devices in order to generate valid quantum conditional probability distributions. A parametrization also allows us to optimize the distribution in order to maximize the rates. However, the bounds on the rates are still device-independent as the SDP is only constrained by the conditional

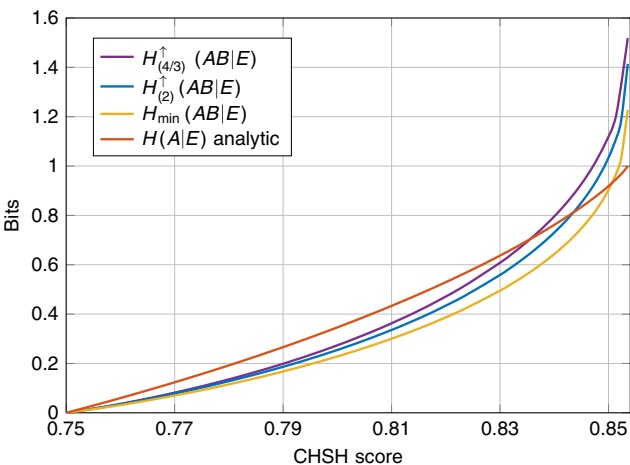

**Fig. 2 Global randomness vs. expected CHSH score.** We compare lower bounds on different measures of the global randomness produced by devices that achieve some minimal expected CHSH score. The curves for $H_{(4/3)}^{\uparrow}(AB|E)$, $H_{(2)}^{\uparrow}(AB|E)$, and $H_{\min}(AB|E)$ were computed numerically, and the red curve representing $\inf H(A|E)$ is computed using the analytical expression from ref. [10].

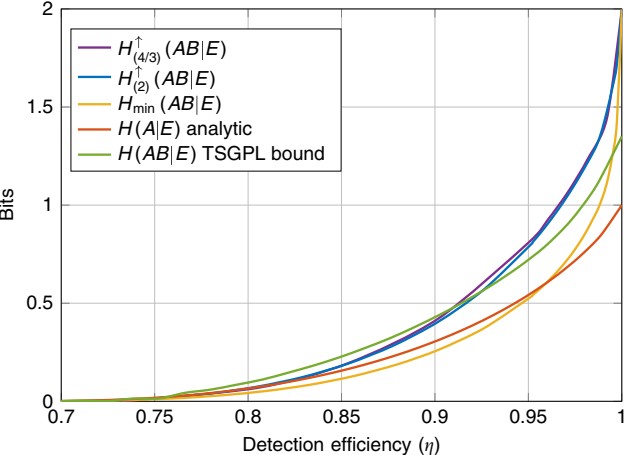

**Fig. 3 Global randomness vs. detection efficiency ($\eta$) in the 2222-scenario.** We compare lower bounds on different measures of the global randomness produced by 2-input 2-output devices that have some fixed detection efficiency $\eta \in [0.7, 1]$. The curves for $H_{(4/3)}^{\uparrow}(AB|E)$, $H_{(2)}^{\uparrow}(AB|E)$, and $H_{\min}(AB|E)$ were computed numerically, the red curve representing $\inf H(A|E)$ was computed using the analytical expression from ref. [10] and the TSGPL bound uses data from the authors of ref. [15]. The red curve (analytic) was computed by maximizing the CHSH score over two-qubit systems with a fixed $\eta$. All other curves constrained the devices to satisfy some fixed probability distribution. For the TSGPL bound this distribution was chosen by maximizing the CHSH score for a fixed $\eta$. For the remainder of the curves we optimized our choice of distribution using the method of ref. [52]. Note that this optimization is important when in the presence of inefficient detectors. For example, if we always use the two-qubit system which achieves Tsirelson's bound for the CHSH game in the noiseless case then we could not certify any entropy for detection efficiencies lower than $\eta \approx 0.83$. However, by allowing ourselves to optimize over partially entangled states we can certify entropy down to detection efficiencies of $\eta \approx 0.67$.

probability distribution and is not concerned with the model used to generate it. For the two-qutrit state, we consider the following family

$$\sin(\theta)\cos(\phi)|00\rangle + \sin(\theta)\sin(\phi)|11\rangle + \cos(\theta)|22\rangle, \quad (19)$$

where $\theta \in [0, \pi]$ and $\phi \in [0, 2\pi]$. Furthermore, we assume that each measurement is a three-outcome projective qutrit measurement and we use the parametrization given in ref. [35]. When considering no-click events, Alice and Bob both record these as the outcome 0, except when Bob receives his key-generating input $y = 2$. When Bob inputs $y = 2$, he no longer records a no-click as the outcome 0 but rather as a new outcome $\perp$. This means that Bob has the potential to receive three or four outcomes whenever he inputs his key-generating input. Retaining this information allows us to reduce $H(A|B, X = 0, Y = 2)$ and further improve the key rate[36].

We consider a DI-QKD protocol with one-way error correction[22]. The asymptotic rate of such a protocol is given by the Devetak–Winter rate[37]

$$H(A|E) - H(A|B). \quad (20)$$

Note that this asymptotic rate does not assume that the adversary acts in an i.i.d. manner. Taking the asymptotic limit of the general finite round DI-QKD rates found in ref. [22] and noting that the optimal one-way error correction leaks approximately $nH(A|B)$ bits in an $n$ round protocol we recover the asymptotic i.i.d. rate[10]. We apply our lower bounds on $H(A|E)$ to compute lower bounds on the asymptotic key rates. We compare our results again with the analytical bound on $H(A|E)$ and numerical bounds on $H_{\min}(A|E)$. The results for devices with two outputs are presented in Fig. 4 and for devices with three outputs in Fig. 5.

Producing high rates in DI-QKD is more difficult than just certifying randomness as the randomness needs to also be correlated between the two devices. In this application, we see an even larger separation between the rates certified by the different entropies. In particular, the minimal detection efficiency required to produce a positive rate differs substantially between the different entropies. In Fig. 4, the curve generated by $H_{\min}$ has a detection efficiency threshold is just below 0.91, for $H_{(2)}^{\uparrow}$ it is just above 0.87, for $H_{(4/3)}^{\uparrow}$ it is just below 0.85, and for $H_{(8/7)}^{\uparrow}$ it is around 0.843. On the inset plot, we zoom in on the region

$[0.84, 0.88] \times [0.0, 0.05]$ and find that the detection efficiency threshold for the protocols based on $H_{(8/7)}^{\uparrow}$ and $H_{(4/3)}^{\uparrow}$ are significantly smaller than the protocol based on the CHSH game. Moreover, the rates certified by $H_{(8/7)}^{\uparrow}$ are larger than those certified by the analytical bound on $H(A|E)$ for all $\eta < 0.92$. Similarly the rates certified by $H_{(4/3)}^{\uparrow}$ are larger than the rates certified by the analytical bound for all $\eta < 0.91$.

For devices with two outputs, the rates are capped at one bit. However, for devices with three possible outputs we see in Fig. 5 that it is possible to achieve a key rate of up to $-\log(1/3) \approx 1.59$ bits. For the curves based on $H_{(4/3)}^{\uparrow}(A|E)$ and $H_{(2)}^{\uparrow}(A|E)$ at around $\eta = 0.97$ and for the curve generated by $H_{\min}(A|E)$ at around $\eta = 0.985$ we see a sharp turn in the rates. This appears to correspond to a transition point where the optimal state found by our optimization transitions from having a Schmidt rank of three to a Schmidt rank of two. Therefore, to the left of these points the optimal strategy found by the optimization could be implemented using a two-qubit system and qubit POVMs. However, when moving from devices with two outputs to three outputs we do not see a significant change in the detection efficiency thresholds. For the curve generated by $H_{\min}$ the threshold is below 0.91. For the curve generated by $H_{(2)}^{\uparrow}$ the threshold is around 0.87 and for $H_{(4/3)}^{\uparrow}$ the threshold detection efficiency is again just below 0.85. From our results it seems that the detection efficiency thresholds do not improve for two-qubit systems by moving from two-outcome protocols to three-outcome protocols. However, this

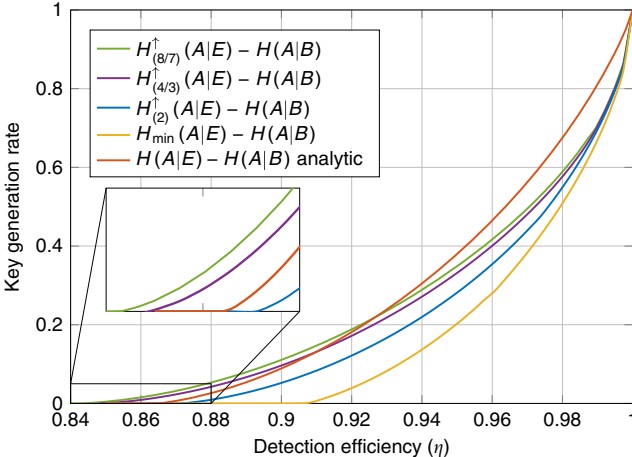

**Fig. 4 Asymptotic key rate vs. detection efficiency ($\eta$) in the 2322-scenario.** We compare lower bounds on the asymptotic key rate for devices that have some fixed detection efficiency $\eta \in [0.84, 1]$. The curves based on the entropies $H^{\uparrow}_{(8/7)}(A|E)$, $H^{\uparrow}_{(4/3)}(A|E)$, $H^{\uparrow}_{(2)}(A|E)$, and $H_{\min}(A|E)$ were computed numerically using the NPA hierarchy, the red curve representing $\inf H(A|E) - H(A|B)$ was computed using the analytical expression from ref. [10]. The curves computed in the NPA hierarchy constrained the devices to satisfy some conditional probability distribution. We optimized the choice of distribution for each detection efficiency and entropy separately using a parametrization of two-qubit systems together with the method of ref. [52].

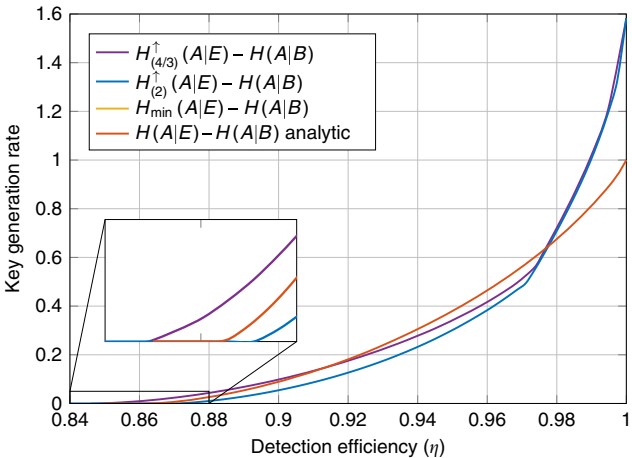

**Fig. 5 Asymptotic key rate vs. detection efficiency ($\eta$) in the 2333-scenario.** We compare lower bounds on the asymptotic key rate for devices that have some fixed detection efficiency $\eta \in [0.84, 1]$. The curves based on the entropies $H^{\uparrow}_{(4/3)}(A|E)$, $H^{\uparrow}_{(2)}(A|E)$, and $H_{\min}(A|E)$ were computed numerically using the NPA hierarchy, the red curve representing $\inf H(A|E) - H(A|B)$ was computed using the analytical expression from ref. [10]. The curves computed in the NPA hierarchy constrained the devices to satisfy some conditional probability distribution. We optimized the choice of distribution for each detection efficiency and entropy separately using a parametrization of two-qutrit systems together with the method of Assad et al.[52].

may also be a consequence of the system optimization finding a local optima. Regardless our results show that we can implement DI-QKD protocols with two-qubit systems with smaller detection efficiencies. The achievability with two-qubit systems is of particular importance to experimental implementations where we seek robust protocols with simple setups.

Using the entropy accumulation theorem[20,21] it would also be possible to calculate explicit lower bounds on the key rates for protocols with a finite number of rounds. In order to apply the EAT we must construct a min-tradeoff function (see Eqs. (11) and (12)). By Lagrangian duality, we can extract from the dual solution to our NPA SDPs, an affine function

$$f : p_{AB|XY} \mapsto \alpha + \sum_{a,b,x,y} \lambda_{abxy} p(a, b|x, y). \qquad (21)$$

We discuss in further detail how to construct min-tradeoff functions from our SDPs and the application to the EAT in the Supplementary Information. For example, let us consider the two-outcome protocols plotted in Fig. 4. For each curve and each value of $\eta$ we searched a two-qubit system to generate a conditional distribution that maximized the rate. Let us take the two-qubit system used for $H^{\uparrow}_{(2)}(A|E)$ at the point $\eta = 0.95$. This system is parameterized by six real numbers ($\theta, a_0, a_1, b_0, b_1, b_2$). The state of the system is $|\psi\rangle = \cos(\theta)|00\rangle + \sin(\theta)|11\rangle$, Alice's measurements are defined by the projectors $M_{0|x} = (I + \cos(a_x)\sigma_z + \sin(a_x)\sigma_x)/2$ and Bob's measurements by the projectors $N_{0|y} = (I + \cos(b_y)\sigma_z + \sin(b_y)\sigma_x)/2$. For this particular system the parameters were $(0.579, -0.161, 1.509, -1.207, 0.660, -0.177)$ and according to the solutions of the optimization we can use it to certify 0.415 bits of entropy and a DI-QKD rate of 0.282 bits when $\eta = 0.95$. Looking at the dual solution we can extract the function

$$
\begin{aligned}
g(p) := &-2\log\left(93.340 - 1.558\, p(0,0|0,0) - 1.599\, p(0,0|0,1)\right.\\
&+ 93.940\, p(0,0|1,0) - 1.709\, p(0,0|1,1) + 1.596\, p_A(0|0)\\
&\left. - 92.340\, p_B(0|0) - 92.334\, p_A(0|1) + 1.706\, p_B(0|1)\right)
\end{aligned}
$$
$$(22)$$

which should lower bound $\inf H(A|E)$. To obtain a min-tradeoff function we can take a first-order Taylor expansion about some distribution, for example, the distribution parametrizing the SDP, which gives us affine lower bounding function[28]. Note that for brevity we have only written the coefficients to three decimal places. As such, this function can likely only guarantee a lower bound up to one or two decimal places, however, this precision can be increased by retaining the precision of the SDP solution.

**Application: Qubit randomness from sequential measurements.** As a final application, we consider the question of how much local entropy can be device-independently certified from a two-qubit system. For example, it is well known that a score of $\cos(\pi/8)^2$ in the CHSH game self tests a maximally entangled two-qubit state[38]. In such a case, the local statistics are uniformly distributed over $\{0, 1\}$ and so this allows us to certify one bit of randomness using a two-qubit system. It has also been shown that up to two bits of local randomness can be certified from a two-qubit system using strategies that include four-outcome qubit POVMs[39].

It is also possible to consider scenarios wherein one party measures multiple times on their half of the two-qubit system. By using unsharp measurements[40] it is possible to measure a two-qubit state such that the post-measurement state remains entangled. Therefore, a two-qubit state can be used to generate multiple instances of nonlocal correlations[41] and in turn a sequence of certifiably random outcomes. The entropy of the sequence of measurement outcomes can then be lower bounded in a device-independent way by using an extension of the NPA hierarchy to sequential correlations[42]. In ref. [42] the authors give an example (ref. [42], Section 4.1) of a two-party scenario in which Bob measures his system twice. They gave an explicit two-

qubit setup, with a state $p|\phi^+\rangle\langle\phi^+| + (1-p)I/4$ where $|\phi^+\rangle = \frac{1}{\sqrt{2}}(|00\rangle + |11\rangle)$ and $p \in [0, 1]$ such that $H_{\min}(B_1B_2|E) > 2$ for a range of $p$. Here $B_1$ refers to the outcome of Bob's first measurement and $B_2$ to his second. As before we look at the entropy only on particular inputs to the devices.

In Fig. 6, we reproduce Fig. 3 from ref. [42] which computes a lower bound on $H_{\min}(B_1B_2|E)$ and compares with the randomness certified by $H_{\min}(A|E)$ for a single two-outcome projective measurement. To illustrate our technique we also include a lower bound on $H_{(2)}^{\uparrow}(B_1B_2|E)$. We see that for low noise the randomness as measured by $H_{(2)}^{\uparrow}(B_1B_2|E)$ can be noticeably larger than $H_{\min}(B_1B_2|E)$. Unlike the previous two examples no concrete protocol or security proof was studied for this scenario and thus neither $H_{(2)}^{\uparrow}(B_1B_2|E)$ nor $H_{\min}(B_1B_2|E)$ correspond to actual rates. However, the example does illustrate that our conditional entropies can also be computed in more exotic scenarios where previously bounds on $H_{\min}$ have been used.

## Discussion
In this work we introduced a new family of Rényi divergences that correspond to convex optimization problems. We showed that the conditional entropies defined by these divergences are amenable to device-independent optimization and can be used as tools to derive numerical lower bounds on the conditional von Neumann entropy. We applied this to the task of computing lower bounds on the rates of device-independent randomness generation and quantum key distribution protocols. We compared the protocol rates derived from our techniques to the analytical bound of refs. [10,22], the numerical techniques of ref. [15], and bounds established via the min-entropy[11–13]. We found improvements over all three of these bounds in various settings.

In particular, when looking at randomness generation in low noise regimes we found improvements over all the previous methods. But in the higher noise regimes, our bounds typically were outperformed by the numerical techniques of ref. [15] in the scenarios where we could compare. However, this comparison has only been performed for some simple protocols where the data for ref. [15] is available. We suspect that our approach is more computationally efficient and thus could be used to analyze a wider range of scenarios. For example, the noncommutative polynomial optimization problems that we evaluated were of degree at most 3 regardless of the number of inputs and outputs of the devices whereas for ref. [15], the degree is six for the smallest possible setting and it grows with the number of inputs and outputs. In addition, the coefficients appearing in the SDPs are explicit small integers for our method whereas for ref. [15], they involve closed-form solutions to integrals of the $\beta$ functions appearing in the multivariate trace inequality of ref. [43]. Finally, our method is flexible to use as it has a parameter $k \in \mathbb{N}$ that can be increased to improve the bounds at the cost of increasing the size of the resulting SDP. The computational efficiency of our method allowed us to iteratively optimize over two-qubit protocols to improve the randomness certification rates up to the maximum of two bits. It is also possible that a combination of the two approaches could yield even higher rates. That is, our techniques could be used to search for optimized protocols and then if the TSGPL bound could be computed it may yield further improvements on the rates.

When computing key rates for DI-QKD, we also looked at bipartite protocols using devices with two and three outputs. There we found significant improvements in the minimal detection efficiency required to generate key in protocols

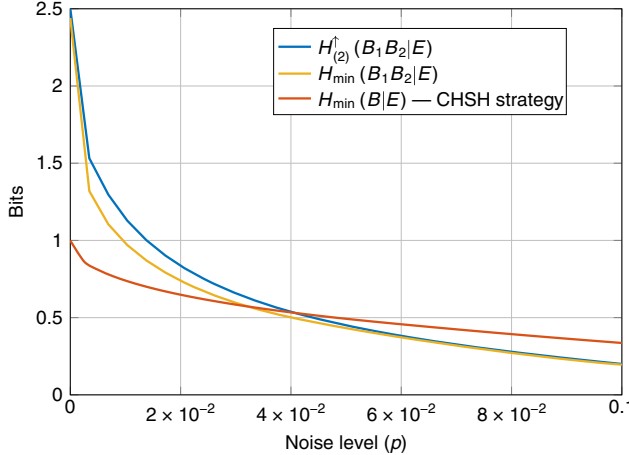

**Fig. 6 Certifiable randomness from sequential measurements.** We compare lower bounds on the certifiable randomness produced by two sequential measurements on one half of the two-qubit state $p|\phi^+\rangle\langle\phi^+| + (1-p)I/4$. All curves are computed numerically using the sequential correlation hierarchy of ref. [42]. This plot extends Fig. 3 from ref. [42] to include the randomness certified by $H_{(2)}^{\uparrow}(B_1B_2|E)$.

without noisy preprocessing. Moreover, in the regimes of higher noise, all of these protocols were still implementable using entangled states of two qubits. It is possible that by further increasing the number of inputs/outputs or by searching for protocols compatible with higher dimensional systems that additional improvements could be made but we leave such an investigation to future work. Reducing the minimal detection efficiency is important for practical experiments, recent works[18,19] have shown that noisy preprocessing of the raw key could also be used to improve minimal detection efficiency for a protocol based on the CHSH game. It would be interesting to see if this could be combined with our numerical techniques to further improve the detection efficiency threshold and design more robust device-independent protocols.

We also demonstrated that min-tradeoff functions could be derived directly from solutions to our device-independent optimizations. These functions can be combined with the entropy accumulation theorem in order to construct simple security proofs for device-independent protocols[22,23]. Therefore, not only can our conditional entropies be used to derive lower bounds on the rates of various protocols but they can also be used directly with the EAT to establish their security proofs and compute finite-key rates. We note that it is not clear if the TSGPL method can be used in the same way.

As a final example, we also showed that our techniques could be used in conjunction with the newly introduced semidefinite hierarchy for sequentially generated quantum correlations[42]. Repeating an example from ref. [42], which looked at the randomness generated from two sequential measurements, we showed higher rate curves could be obtained by using $H_{(2)}^{\uparrow}$ as opposed to $H_{\min}$ which was the measure of randomness originally used in the example.

Several additional questions remain open from this work. First, what can be said about the limit $D_{(\alpha_k)}$ as $k \to \infty$? We know that it will be between the Umegaki divergence $D$ and the Belavkin–Staszewski divergence $\widehat{D}$, and we also know that it cannot always be equal to $\widehat{D}$ (there are some examples where already $D_{(2)} < \widehat{D}$). If one can show that $\lim_{k\to\infty} D_{(\alpha_k)} = D$, then this shows that our technique can approximate the conditional von Neumann entropy arbitrarily well. However, certain

numerical evidence indicates that if these quantities do converge to $D$ then the convergence will be slow (see Fig. 2 in the Supplementary Information for an example). More generally, a very interesting question is whether other divergences that provide a tighter approximation to the Umegaki divergence (e.g., the sandwiched Rényi divergence) have a free variational expression as discussed in Remark 1.

Second, it would be interesting to see whether our computations can be made more efficient. For example, we know that there exists a particular dilation theorem can be applied to reduce the size of the optimization $H^{\uparrow}_{(2)}$ and speed up its convergence (see Supplementary Information). It would be interesting to see whether one could extend this dilation theorem to other $H^{\uparrow}_{(\alpha_k)}$ to help improve their convergence and efficiency also. In addition, it may be possible to reduce the size of the $H^{\uparrow}_{(\alpha_k)}$ optimizations by exploiting symmetries of the problem[44] or by optimizing the choice of monomial sets generating the NPA moment matrices.

## Methods

Here we state and prove some of the main properties of the iterated mean divergences defined in Definition 1. Note that we refer to them as quantum Rényi divergences as they match for commuting operators with the classical Rényi divergence. As Rényi divergences are well-studied objects in information theory and have found numerous operational interpretations the iterated mean divergences may also be of independent interest. In fact, our new divergences have already inspired the definition of other quantum divergences with different information-theoretic applications[45]. The key property of iterated mean divergences that makes them suited for DI optimization is that their SDP representation does not explicitly refer to the dimension of the underlying quantum systems. With this property, the corresponding conditional entropy of a state $\rho$ can be written as a maximization of a noncommutative Hermitian polynomial in some operators $V_1, \ldots, V_m$ evaluated on the state $\rho$ and the operators $V_1, \ldots, V_m$ are subject to polynomial inequalities that are dimension independent. We refer to Remark 1 for a more detailed discussion of this point.

**Preliminaries.** We define $\mathbb{N}$ to be the set of strictly positive integers. Let $\mathcal{H}$ be a Hilbert space; we denote the set of linear operators on $\mathcal{H}$ by $\mathcal{L}(\mathcal{H})$, the set of Hermitian operators on $\mathcal{H}$ by $\mathcal{H}(\mathcal{H})$, the set of positive-semidefinite operators on $\mathcal{H}$ by $\mathcal{P}(\mathcal{H})$ and the set of positive-semidefinite operators with unit trace on $\mathcal{H}$ by $\mathcal{D}(\mathcal{H})$. All Hilbert spaces in this work are finite-dimensional unless otherwise stated. Given a linear map $\mathcal{E}: \mathcal{L}(\mathcal{H}_1) \to \mathcal{L}(\mathcal{H}_2)$, we say $\mathcal{E}$ is CPTP if it is completely positive and trace preserving. Given two Hilbert spaces $\mathcal{H}$ and $\mathcal{K}$ we write $\mathcal{H}\mathcal{K}$ as shorthand for $\mathcal{H} \otimes \mathcal{K}$. Given two operators $A, B \in \mathcal{L}(\mathcal{H})$ we write $A \leq B$ if $B - A \in \mathcal{P}(\mathcal{H})$. The support of an operator $A \in \mathcal{L}(\mathcal{H})$, denoted $\text{supp}(A)$, is the orthogonal complement of its kernel, $\ker(A) = \{x \in \mathcal{H} : Ax = 0\}$. For $A, B \in \mathcal{L}(\mathcal{H})$, we write $A \subseteq B$ if $\text{supp}(A) \subseteq \text{supp}(B)$. For $A \in \mathcal{L}(\mathcal{H})$, $A^*$ denotes its adjoint and if $A$ is nonsingular then $A^{-1}$ denotes its inverse. If $A$ is singular then $A^{-1}$ denotes the Moore–Penrose pseudo-inverse of $A$. We use the symbol $I$ to denote the identity operator. A collection of operators $\{M_1, \ldots, M_n\}$ forms an $n$-outcome POVM on $\mathcal{H}$ if $\sum_{i=1}^{n} M_i = I$ and $M_i \in \mathcal{P}(\mathcal{H})$ for all $i = 1, \ldots, n$. Throughout this work, we shall be interested in classical systems that arise from measurements on some quantum system. To distinguish the classical and quantum systems in such a measurement process we shall often write a single uppercase Roman character to denote the classical system resulting from the measurement, e.g., $A$, and denote the corresponding quantum system from which it is obtained by $Q_A$.

The geometric mean of two positive definite matrices $A$ and $B$ is defined as

$$A\#B = A^{1/2}\left(A^{-1/2}BA^{-1/2}\right)^{1/2}A^{1/2}.$$

This definition can be extended to positive-semidefinite matrices $A$, $B$ as $\lim_{\epsilon\to 0} A_\epsilon \# B_\epsilon$ where $X_\epsilon = X + \epsilon I$. The geometric mean has the property that if $C \leq D$ then $A\#C \leq A\#D$ (ref. [46], Corollary 3.2.3).

Let $\alpha \in (0, 1) \cup (1, \infty)$, $\rho \in \mathcal{D}(\mathcal{H})$ and $\sigma \in \mathcal{P}(\mathcal{H})$ with $\rho \ll \sigma$. The Petz–Rényi divergence[47] of order $\alpha$ is defined as

$$\overline{D}_\alpha(\rho \| \sigma) := \frac{1}{\alpha - 1} \log \text{Tr}\left[\rho^\alpha \sigma^{1-\alpha}\right]. \tag{23}$$

The sandwiched Rényi divergence[26,27] of order $\alpha$ is defined as

$$\widetilde{D}_\alpha(\rho \| \sigma) := \frac{1}{\alpha - 1} \log \text{Tr}\left[\left(\sigma^{\frac{1-\alpha}{2\alpha}} \rho \sigma^{\frac{1-\alpha}{2\alpha}}\right)^\alpha\right]. \tag{24}$$

In the limit $\alpha \to 1$ both the Petz–Rényi divergence and the sandwiched Rényi

divergence converge to the Umegaki relative entropy[48]

$$D(\rho \| \sigma) := \text{Tr}[\rho(\log \rho - \log \sigma)]. \tag{25}$$

The geometric Rényi divergence[49] of order $\alpha$ is defined as

$$\widehat{D}_\alpha(\rho \| \sigma) := \frac{1}{\alpha - 1} \log \text{Tr}\left[\rho^{1/2}\left(\rho^{-1/2}\sigma\rho^{-1/2}\right)^{1-\alpha}\rho^{1/2}\right]. \tag{26}$$

In the limit $\alpha \to 1$ the geometric Rényi divergence converges to the Belavkin–Staszewski relative entropy $\text{Tr}\left[\rho\log\left(\rho^{1/2}\sigma^{-1}\rho^{1/2}\right)\right]$[50]. The geometric Rényi divergence is the largest Rényi divergence satisfying data processing. The max divergence is defined as

$$D_{\max}(\rho \| \sigma) := \log \inf\{\lambda > 0 : \rho \leq \lambda\sigma\}. \tag{27}$$

Finally, the measured Rényi divergence is defined as the largest classical divergence obtained from measuring $\rho$ and $\sigma$. For $\alpha \in (1, \infty)$ this is formally defined as

$$D_\alpha^{\mathbb{M}}(\rho \| \sigma) := \frac{1}{\alpha - 1} \log \sup_{\{M_i\}_i} \sum_i \text{Tr}[M_i\rho]^\alpha \text{Tr}[M_i\sigma]^{1-\alpha}, \tag{28}$$

where the supremum is taken over all POVMs $\{M_i\}$. This divergence also admits the following variational characterization[51]

$$D_\alpha^{\mathbb{M}}(\rho \| \sigma) = \frac{1}{\alpha - 1} \log \sup_{\omega > 0} \alpha\text{Tr}\left[\rho\omega^{1-\frac{1}{\alpha}}\right] + (1 - \alpha)\text{Tr}[\sigma\omega]. \tag{29}$$

Given bipartite state $\rho_{AB} \in \mathcal{D}(AB)$ and a Rényi divergence $\mathbb{D}$ we define a corresponding conditional entropy

$$\mathbb{H}^{\downarrow}(A|B)_\rho := -\mathbb{D}(\rho_{AB} \| I_A \otimes \rho_B) \tag{30}$$

and a corresponding optimized conditional entropy

$$\mathbb{H}^{\uparrow}(A|B)_\rho := \sup_{\sigma_B \in \mathcal{D}(B)} -\mathbb{D}(\rho_{AB} \| I_A \otimes \sigma_B). \tag{31}$$

The min-entropy is defined as

$$H_{\min}(A|B) = \sup_{\sigma_B \in \mathcal{D}(B)} -D_{\max}(\rho_{AB} \| I_A \otimes \sigma_B). \tag{32}$$

**Properties of the iterated mean divergences.** The following proposition details some alternate formulations and properties of the iterated mean divergences. We defer the proof of this proposition to the Supplementary Information.

**Proposition 2.** Let $\rho \in \mathcal{D}(\mathcal{H})$, $\sigma \in \mathcal{P}(\mathcal{H})$ and $k \in \mathbb{N}$. Then the following all hold:

1. (Rescaling)

$$
\begin{aligned}
Q_{(\alpha_k)}(\rho \| \sigma) = \max_{V_1, \ldots, V_k, Z} \quad & \left(\text{Tr}\left[\rho\frac{(V_1 + V_1^*)}{2}\right]\right)^{\alpha_k} \\
\text{s.t.} \quad & \text{Tr}[\sigma Z] = 1 \\
& V_1 + V_1^* \geq 0 \\
& \begin{pmatrix} I & V_i \\ V_i^* & \frac{(V_{i+1} + V_{i+1}^*)}{2} \end{pmatrix} \geq 0 \quad \text{for } 1 \leq i \leq k-1 \\
& \begin{pmatrix} I & V_k \\ V_k^* & Z \end{pmatrix} \geq 0.
\end{aligned}
\tag{33}
$$

2. (Dual formulations) We have

$$
\begin{aligned}
Q_{(\alpha_k)}(\rho \| \sigma) = \min_{A_1, \ldots, A_k, C_1, \ldots, C_k} \quad & \frac{1}{2^k - 1}\sum_{i=1}^{k} 2^{k-i}\text{Tr}[A_i] \\
\text{s.t.} \quad & C_1 \geq \rho \\
& \begin{pmatrix} A_i & C_i \\ C_i & C_{i+1} \end{pmatrix} \geq 0 \quad \text{for } 1 \leq i \leq k-1 \\
& \begin{pmatrix} A_k & C_k \\ C_k & \sigma \end{pmatrix} \geq 0.
\end{aligned}
\tag{34}
$$

Or also

$$
\begin{aligned}
Q_{(\alpha_k)}(\rho \| \sigma) = \min_{A_1, \ldots, A_k, C_1, \ldots, C_k} \quad & \text{Tr}[A_1] \\
\text{s.t.} \quad & C_1 \geq \rho \\
& \text{Tr}[A_1] = \text{Tr}[A_2] = \cdots = \text{Tr}[A_k] \\
& \begin{pmatrix} A_i & C_i \\ C_i & C_{i+1} \end{pmatrix} \geq 0 \quad \text{for } 1 \leq i \leq k-1 \\
& \begin{pmatrix} A_k & C_k \\ C_k & \sigma \end{pmatrix} \geq 0.
\end{aligned}
\tag{35}
$$

Finally and eponymously

$$Q_{(\alpha_k)}(\rho\|\sigma) = \min_{A_1,\dots,A_k} \ \mathrm{Tr}[A_1]$$
$$\text{s.t.} \quad \mathrm{Tr}[A_1] = \mathrm{Tr}[A_2] = \cdots = \mathrm{Tr}[A_k] \tag{36}$$
$$\rho \le A_1 \# (A_2 \# (\dots \# (A_k \# \sigma) \dots )).$$

3. (Submultiplicativity) Let $\rho_1 \in \mathscr{D}(\mathcal{H}_1)$, $\sigma_1 \in \mathscr{P}(\mathcal{H}_1)$, $\rho_2 \in \mathscr{D}(\mathcal{H}_2)$ and $\sigma_2 \in \mathscr{P}(\mathcal{H}_2)$. Then,

$$D_{(\alpha_k)}(\rho_1 \otimes \rho_2 \| \sigma_1 \otimes \sigma_2) \le D_{(\alpha_k)}(\rho_1 \| \sigma_1) + D_{(\alpha_k)}(\rho_2 \| \sigma_2). \tag{37}$$

4. (Relation to other Rényi divergences)

$$D_{\alpha_k}^{\mathbb{M}}(\rho\|\sigma) \le \widetilde{D}_{\alpha_k}(\rho\|\sigma) \le D_{(\alpha_k)}(\rho\|\sigma) \le \widehat{D}_{\alpha_k}(\rho\|\sigma) \tag{38}$$

5. (Decreasing in k) For all $k \ge 2$,

$$D_{(\alpha_k)}(\rho\|\sigma) \le D_{(\alpha_{k-1})}(\rho\|\sigma). \tag{39}$$

6. (Data processing) Let $\mathcal{K}$ be another Hilbert space and let $\mathcal{E}: \mathscr{L}(\mathcal{H}) \to \mathscr{L}(\mathcal{K})$ be a CPTP map, then

$$D_{(\alpha_k)}(\rho\|\sigma) \ge D_{(\alpha_k)}(\mathcal{E}(\rho)\|\mathcal{E}(\sigma)). \tag{40}$$

7. (Reduction to classical divergence) If $[\rho,\sigma]=0$ then

$$D_{(\alpha_k)}(\rho\|\sigma) = \frac{1}{\alpha_k - 1}\log \mathrm{Tr}\left[\rho^{\alpha_k}\sigma^{1-\alpha_k}\right]. \tag{41}$$

Remark 2 (Relation to $\overline{D}_2(\rho\|\sigma)$). We can show that $D_{(2)}(\rho\|\sigma)$ is no larger than the Petz–Rényi divergence $\overline{D}_2(\rho\|\sigma)$. Note that we have $\begin{pmatrix} A & B \\ B^* & C \end{pmatrix} \ge 0 \iff$ $C \ge 0$, $(I - CC^{-1})B^* = 0$ and $A \ge BC^{-1}B^*$. Applying this identity to the second dual form (34) we find the optimal choice for the $A_i$ variables is $A_i = C_i C_{i+1}^{-1} C_i$ for $1 \le i \le k-1$ and $A_k = C_k \sigma^{-1} C_k$. For this particular choice, the objective function becomes

$$\sum_{i=1}^{k-1} \frac{2^{k-i}}{2^k - 1}\mathrm{Tr}\left[C_i^2 C_{i+1}^{-1}\right] + \frac{1}{2^k-1}\mathrm{Tr}\left[C_k^2 \sigma^{-1}\right]. \tag{42}$$

This expression is a convex combination of terms of the form $\overline{Q}_2(A\|B) = \mathrm{Tr}[A^2 B^{-1}]$, i.e., the Petz generalized mean of order 2. We see for $\alpha_k = 2$ the problem reduces to

$$\min_{C_1} \ \mathrm{Tr}\left[C_1^2 \sigma^{-1}\right]$$
$$\text{s.t.} \quad C_1 \ge \rho. \tag{43}$$

For the feasible point $C_1 = \rho$ we recover $\overline{Q}_2(\rho\|\sigma) = \mathrm{Tr}[\rho^2\sigma^{-1}]$ and so $\overline{Q}_2(\rho\|\sigma) \ge Q_{(2)}(\rho\|\sigma)$ and therefore by monotonicity of the logarithm $\overline{D}_2(\rho\|\sigma) \ge D_{(2)}(\rho\|\sigma)$. Furthermore, if we drop the constraint $V_1 + V_1^* \ge 0$ from the definition of $D_{(2)}(\rho\|\sigma)$ then one can show that $D_{(2)}(\rho\|\sigma) = \overline{D}_2(\rho\|\sigma)$.

**Proof of Proposition 1.** We now provide a proof of Proposition 1 which gives a variational characterization of $H^\uparrow_{(\alpha_k)}(A|B)$. More explicitly, it states for a bipartite state $\rho \in \mathscr{D}(AB)$ we can write

$$H^\uparrow_{(\alpha_k)}(A|B)_\rho = \frac{1}{1-\alpha_k}\log Q^\uparrow_{(\alpha_k)}(\rho) \tag{44}$$

where

$$Q^\uparrow_{(\alpha_k)}(\rho) = \max_{V_1,\dots,V_k} \ \left(\mathrm{Tr}\left[\rho\frac{(V_1+V_1^*)}{2}\right]\right)^{\alpha_k}$$
$$\text{s.t.} \quad \mathrm{Tr}_A\left[V_k^* V_k\right] \le I_B$$
$$V_1 + V_1^* \ge 0 \tag{45}$$
$$\begin{pmatrix} I & V_i \\ V_i^* & \frac{(V_{i+1}+V_{i+1}^*)}{2} \end{pmatrix} \ge 0 \quad \text{for } 1 \le i \le k-1.$$

Proof. By the definition of $H^\uparrow_{(\alpha_k)}(A|B)$ we have $H^\uparrow_{(\alpha_k)}(A|B) = \sup_{\sigma_B} -D_{(\alpha_k)}(\rho_{AB}\|I_A\otimes\sigma_B)$ which in turn is equal to

$$\frac{1}{1-\alpha_k}\log\inf_{\sigma_B}\max_{V_1,\dots,V_k,Z} \ \alpha_k\mathrm{Tr}\left[\rho\frac{(V_1+V_1^*)}{2}\right] - (\alpha_k-1)\mathrm{Tr}[(I_A\otimes\sigma_B)Z]$$
$$\text{s.t.} \quad V_1 + V_1^* \ge 0$$
$$\begin{pmatrix} I & V_i \\ V_i^* & \frac{(V_{i+1}+V_{i+1}^*)}{2} \end{pmatrix} \ge 0 \quad \text{for } 1 \le i \le k-1$$
$$\begin{pmatrix} I & V_k \\ V_k^* & Z \end{pmatrix} \ge 0.$$

Now consider the set $\mathscr{M}$ of points $(V_1,\dots,V_k,Z) \in \mathscr{L}(AB)^{k+1}$ such that $(V_1,\dots,V_k,Z)$ is a feasible point of the above optimization and the function

$f : \mathscr{D}(B) \times \mathscr{M} \to \mathbb{R}$ defined as

$$f(\sigma_B, V_1, \dots, V_k, Z) = \alpha_k\mathrm{Tr}\left[\rho\frac{(V_1+V_1^*)}{2}\right] - (\alpha_k-1)\mathrm{Tr}[(I_A\otimes\sigma_B)Z].$$

Note that $\mathscr{M}$ is a convex set, $\mathscr{D}(B)$ is a compact and convex set and $f$ is a continuous function. In addition, $f$ is both convex and concave in each argument—treating $(V_1,\dots,V_k,Z)$ as one argument. Now we have

$$\inf_{\sigma_B}\max_{V_1,\dots,V_k,Z} f(\sigma_B,V_1,\dots,V_k,Z) \ge \max_{V_1,\dots,V_k,Z}\inf_{\sigma_B} f(\sigma_B,V_1,\dots,V_k,Z)$$
$$= \max_{V_1,\dots,V_k,Z}\min_{\sigma_B} f(\sigma_B,V_1,\dots,V_k,Z)$$
$$= \min_{\sigma_B}\max_{V_1,\dots,V_k,Z} f(\sigma_B,V_1,\dots,V_k,Z)$$
$$\ge \inf_{\sigma_B}\max_{V_1,\dots,V_k,Z} f(\sigma_B,V_1,\dots,V_k,Z)$$

where the second line follows from the fact that $\mathscr{D}(B)$ is compact and $f$ is continuous on $\mathscr{D}(B)$ and the third line from Sion's minimax theorem. Thus, we have

$$\inf_{\sigma_B}\max_{V_1,\dots,V_k,Z} f(\sigma_B,V_1,\dots,V_k,Z) = \max_{V_1,\dots,V_k,Z}\min_{\sigma_B} f(\sigma_B,V_1,\dots,V_k,Z)$$

and so we can interchange the inf max in our optimization for a max min. Now as $\max_{\sigma_B}\mathrm{Tr}[(I_A\otimes\sigma_B)Z] = \lambda_{\max}(\mathrm{Tr}_A[Z])$ we can write $H^\uparrow_{(\alpha_k)}(A|B)$ as

$$\frac{1}{1-\alpha_k}\log\max_{V_1,\dots,V_k,Z} \ \alpha_k\mathrm{Tr}\left[\rho\frac{(V_1+V_1^*)}{2}\right] - (\alpha_k-1)\lambda_{\max}(\mathrm{Tr}_A[Z])$$
$$\text{s.t.} \quad V_1 + V_1^* \ge 0$$
$$\begin{pmatrix} I & V_i \\ V_i^* & \frac{(V_{i+1}+V_{i+1}^*)}{2} \end{pmatrix} \ge 0 \quad \text{for } 1 \le i \le k-1$$
$$\begin{pmatrix} I & V_k \\ V_k^* & Z \end{pmatrix} \ge 0.$$

Finally, by applying the same rescaling arguments used in the proof of property 1 in Proposition 2 we can homogenize the objective function to remove the second term and add the constraint $\lambda_{\max}(\mathrm{Tr}_A[Z]) = 1$. After doing so we arrive at the expression

$$Q^\uparrow_{(\alpha_k)}(\rho) = \max_{V_1,\dots,V_k,Z} \ \left(\mathrm{Tr}\left[\rho\frac{(V_1+V_1^*)}{2}\right]\right)^{\alpha_k}$$
$$\text{s.t.} \quad \lambda_{\max}(\mathrm{Tr}_A[Z]) = 1$$
$$V_1 + V_1^* \ge 0$$
$$\begin{pmatrix} I & V_i \\ V_i^* & \frac{(V_{i+1}+V_{i+1}^*)}{2} \end{pmatrix} \ge 0 \quad \text{for } 1 \le i \le k-1$$
$$\begin{pmatrix} I & V_k \\ V_k^* & Z \end{pmatrix} \ge 0. \tag{46}$$

To derive the second expression we first note that the final positive-semidefinite constraint in Eq. (46) is equivalent to the operator inequality $Z \ge V_k^* V_k$. This condition, together with the fact that $V_k^* V_k \ge 0$, implies that $1 = \lambda_{\max}(\mathrm{Tr}_A[Z]) \ge \lambda_{\max}(\mathrm{Tr}_A[V_k^* V_k]) \ge 0$. Now notice that for any feasible point $(V_1,\dots,V_k,Z)$ of Eq. (46), the point $\left(V_1,\dots,V_k,\frac{V_k^* V_k}{\lambda_{\max}(\mathrm{Tr}_A[V_k^* V_k])}\right)$ is also feasible and has the same objective value, we may therefore restrict our consideration to feasible points of this latter form. Furthermore, we have $\lambda_{\max}(\mathrm{Tr}_A[V_k^* V_k]) \le 1 \iff \mathrm{Tr}_A[V_k^* V_k] \le I_B$. We now have a bijection between feasible points $(V_1,\dots,V_k)$ of Eq. (6) and feasible points $\left(V_1,\dots,V_k,\frac{V_k^* V_k}{\lambda_{\max}(\mathrm{Tr}_A[V_k^* V_k])}\right)$ of Eq. (46) which preserves objective values and therefore the two optimizations are equivalent and the proof is complete.

Remark 3 (Relation to $H_{\min}(A|B)$). In Proposition 2 (see Eq. (39)) it was shown via an application of the Cauchy–Schwarz inequality that $D_{(\alpha_k)}(\rho\|\sigma) \le D_{(\alpha_{k-1})}(\rho\|\sigma)$, which in turn implies $H^\uparrow_{(\alpha_k)}(A|B) \ge H^\uparrow_{(\alpha_{k-1})}(A|B)$. Applying the Cauchy–Schwarz inequality to the objective function of $H^\uparrow_{(2)}(A|B)$ we see that

$$-2\log\max_{V_1}\mathrm{Tr}[\rho(V_1+V_1^*)/2] \le -2\log\max_{V_1}\mathrm{Tr}\left[\rho V_1^* V_1\right]^{1/2}$$
$$= -\log\max_{V_1}\mathrm{Tr}\left[\rho V_1^* V_1\right].$$

Therefore we have

$$H^\uparrow_{(2)}(A|B) \ge -\log\max \quad \mathrm{Tr}\left[\rho V_1^* V_1\right]$$
$$\text{s.t.} \quad \mathrm{Tr}_A\left[V_1^* V_1\right] \le I_B$$
$$V_1^* + V_1 \ge 0.$$

Let us compare this optimization with the min-entropy

$$H_{\min}(A|B) = -\log \max_{M \geq 0} \mathrm{Tr}[\rho M]$$
$$\text{s.t} \quad \mathrm{Tr}_A[M] \leq I_B.$$

As $V_1^* V_1 \geq 0$ we see that for each feasible point $V_1$ of the first optimization $V_1^* V_1$ is a feasible point of the second optimization with the same objective value. Conversely, for any feasible point $M$ of the second optimization, $V_1 = M^{1/2}$ is a feasible point of the first optimization with the same objective value and so $H_{(2)}^{\uparrow}(A|B) \geq H_{\min}(A|B)$. Thus, the sequence of conditional entropies $H_{(\alpha_k)}^{\uparrow}(A|B)$ are each separated by a Cauchy–Schwarz inequality and first term $H_{(2)}^{\uparrow}(A|B)$ is separated by another application of the Cauchy–Schwarz inequality from $H_{\min}(A|B)$.

**Proof of Lemma 1.** We now state a proof of Lemma 1 which gave an expression for $H_{(\alpha_k)}^{\uparrow}(A|E)$ where $A$ is a classical system resulting from some measurement on a quantum system $Q_A$ and $E$ is a quantum system that may have been entangled with $Q_A$. The form that $H_{(\alpha_k)}^{\uparrow}(A|E)$ takes in the lemma allowed us to optimize the quantity in a device-independent manner using the NPA hierarchy.

*Proof.* From Proposition 1 we know that we can write $H_{(\alpha_k)}^{\uparrow}(A|E)$ as

$$\frac{\alpha_k}{1-\alpha_k} \log \max_{V_1, \ldots, V_k} \mathrm{Tr}\left[\rho_{AE} \frac{(V_1 + V_1^*)}{2}\right]$$
$$\text{s.t.} \quad \mathrm{Tr}_A\left[V_k^* V_k\right] \leq I_E$$
$$V_1 + V_1^* \geq 0$$
$$\begin{pmatrix} I & V_i \\ V_i^* & \frac{(V_{i+1} + V_{i+1}^*)}{2} \end{pmatrix} \geq 0 \quad \text{for } 1 \leq i \leq k-1.$$

For $1 \leq i \leq k$ let $V_i = \sum_{a,b} |a\rangle\langle b| \otimes \hat{V}_i(a,b)$ for some $\hat{V}_i(a,b) \in \mathscr{L}(E)$. Taking the partial trace over $A$ in the objective function we can rewrite it as

$$\mathrm{Tr}\left[\frac{V_1 + V_1^*}{2} \rho_{AE}\right] = \sum_a \mathrm{Tr}\left[\frac{\hat{V}_1(a,a) + \hat{V}_1^*(a,a)}{2} \rho_E(a)\right]$$
$$= \sum_a \mathrm{Tr}\left[\frac{\hat{V}_1(a,a) + \hat{V}_1^*(a,a)}{2} \mathrm{Tr}_{Q_A}[(M_a \otimes I)|\psi\rangle\langle\psi|]\right]$$
$$= \sum_a \mathrm{Tr}\left[\mathrm{Tr}_{Q_A}\left[\left(M_a \otimes \frac{\hat{V}_1(a,a) + \hat{V}_1^*(a,a)}{2}\right)|\psi\rangle\langle\psi|\right]\right]$$
$$= \sum_a \mathrm{Tr}\left[\left(M_a \otimes \frac{\hat{V}_1(a,a) + \hat{V}_1^*(a,a)}{2}\right)|\psi\rangle\langle\psi|\right].$$

Now for a linear operator $X = \sum_{a,b} |a\rangle\langle b| \otimes X(a,b)$ acting on $AE$ consider the pinching map defined by the action $\mathcal{P}(X) = \sum_a |a\rangle\langle a| \otimes X(a,a)$ that pinches in the classical basis of $A$ defined by the cq-state $\rho_{AE}$. Note that $\mathcal{P}$ is both CP and unital and so it preserves the semidefinite constraints, i.e.,

$$\begin{pmatrix} I & V_i \\ V_i^* & \frac{(V_{i+1} + V_{i+1}^*)}{2} \end{pmatrix} \geq 0 \Rightarrow \begin{pmatrix} I & \mathcal{P}(V_i) \\ \mathcal{P}(V_i)^* & \frac{(\mathcal{P}(V_{i+1}) + \mathcal{P}(V_{i+1})^*)}{2} \end{pmatrix} \geq 0$$

and $V_1 + V_1^* \geq 0 \Rightarrow \mathcal{P}(V_1) + \mathcal{P}(V_1)^* \geq 0$. Furthermore, the variable $W_k = \mathcal{P}(V_k)$ also satisfies the constraint $\mathrm{Tr}_A\left[W_k^* W_k\right] \leq I$ as

$$\mathrm{Tr}_A\left[W_k^* W_k\right] = \sum_a \hat{V}_k^*(a,a)\hat{V}_k(a,a)$$
$$\leq \sum_{a,b} \hat{V}_k^*(a,b)\hat{V}_k(a,b)$$
$$= \mathrm{Tr}_A\left[V_k^* V_k\right]$$
$$\leq I.$$

Finally, the objective function is invariant under the pinching as it only contains block diagonal elements $\hat{V}_k(a,a)$. As such, for any feasible point of the optimization problem, we can replace the variables with their respective pinchings to obtain another feasible point with the same objective function value. We may therefore restrict all variables in the optimization to take the form $V_i = \sum_a |a\rangle\langle a| \otimes \hat{V}_i(a,a)$. Applying the Schur complement lemma (see the Supplementary Information) to the remaining block positive-semidefinite constraints we can rewrite them as operator inequalities. Then by relabeling $\hat{V}_i(a,a)$ to $V_{i,a}$, the result follows. □

With Lemma 1 we can then consider device-independent optimizations of $H_{(\alpha_k)}^{\uparrow}$. That is, we impose some statistical constraints on the classical system(s), e.g., a Bell-inequality violation, and we take the infimum of $H_{(\alpha_k)}^{\uparrow}(A|E)$ over all possible states and measurements on finite-dimensional Hilbert spaces. Using the form of $H_{(\alpha_k)}^{\uparrow}$ given in Lemma 1, the resulting problem is a noncommutative polynomial optimization which can be lower bounded by an SDP using the NPA hierarchy. Much like we can use a Naimark dilation to assume the operators in a device-

independent optimization of $H_{\min}$ are projections, we also found an analogous dilation can be made for $H_{(2)}^{\uparrow}$ under certain restrictions.

*Remark 4.* When $k = 1$ (i.e., $\alpha_k = 2$) and we are optimizing in the device-independent setting we may impose some additional constraints on the operators $\{V_{1,a}\}_a$. Namely, we may assume that for all $a, b \in \mathcal{A}$,

$$V_{1,a} V_{1,b}^* = \delta_{ab} I. \quad (47)$$

This allows us to remove certain monomials from the moment matrix of the relaxed problem, which makes the size of the SDP smaller. Moreover, this implies that the operators $V_{1,a}^* V_{1,a}$ form a set of orthogonal projections. As was shown in Remark 3, we can recover $H_{\min}(A|E)$ from $H_{(2)}^{\uparrow}(A|E)$ by an appropriate application of the Cauchy–Schwarz inequality. In that case the operators $\{V_a^* V_a\}_{a \in \mathcal{A}}$ played the role of Eve's POVM $\{W_a\}_{a \in \mathcal{A}}$. By adding the additional constraints (47) to the optimization problem defining $H_{(2)}^{\uparrow}(A|E)$ the operators $\{V_a^* V_a\}_{a \in \mathcal{A}}$ now form a projective measurement. This can be an important additional constraint as imposing that measurements are projective when computing $H_{\min}(A|E)$ often speeds up its convergence in the NPA hierarchy. Thus, the constraints (47) can also be helpful in this regard. However, in order to impose these constraints, we have to remove or relax the constraint $V_1 + V_1^* \geq 0$. In practice, when computing the rates in the proceeding sections, we remove the constraint $V_1 + V_1^* \geq 0$ as we did not observe any change as a result. It is also possible to include these additional constraints in the optimizations of $H_{(\alpha_k)}^{\uparrow}$ for $k \geq 2$. However, this requires additional considerations. We discuss this further in the Supplementary Information.

## Data availability
The datasets used in this work are available from the corresponding author upon reasonable request.

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

## Acknowledgements

This research is supported by the French National Research Agency via Project No. ANR-18-CE47-0011 (ACOM). P.B. thanks Ernest Tan for useful discussions and for providing data used in the comparisons with the TSGPL bound. P.B. also thanks Joseph Bowles for providing code used for generating the semidefinite relaxations in the sequential correlation example.

## Author contributions

P.B., H.F., and O.F. contributed equally to this work.

## Competing interests

The authors declare no competing interests.
