## [Peer Review File · Nature Communications]

Response to reviewer comments

We are grateful to both reviewers for taking the time to comment on our manuscript. We include our reply to their comments below.

Reviewer 1

A quantum information protocol is said to be device-independent if its soundness can be certified through Bell-type experiments, without relying on a technical description of the measurement and preparation apparatuses involved. In this regard, device-independent quantum key distribution and randomness generation are the most practical applications. The former allows to establish a one-time pad between two separate parties; the latter, to generate a secret string of random bits. Proving security in either case requires lower bounding conditional entropies such as $H(AB|X = x, Y = y, E)$ from the observed correlations $P(A, B|X, Y)$. Here X, Y denote the local inputs of each party; and A, B their respective outcomes. E represents a quantum system held by an eavesdropper.

The traditional way of establishing a lower bound on such entropic quantities starts by approximating the corresponding figure by the min-entropy, that quantifies how well the eavesdropper can guess the outcome of a measurement. This, in turn, can be expressed as an optimization over the set of feasible quantum correlations, a problem that can be tackled with the NPA hierarchy. The min-entropy is, however, a strict lower bound on the desired quantity; results relying on min-entropies thus risk overestimating the power of the eavesdropping party.

In their manuscript, Brown et al. introduce what they call iterated mean divergences, a decreasing infinite sequence of functions of the quantum state which, like the min-entropy, allow one to lower bound the relevant conditional entropies. These divergences have two very useful properties, namely: a) in general, they provide better bounds than the min-entropy; b) they can be expressed as a semidefinite program (SDP) with matrix constraints independent of the Hilbert space dimension where the quantum state acts.

This last property allows the authors to formulate the estimation of the iterated mean divergence in a Bell experiment as a non-commutative polynomial optimization problem. The authors use NPA-like tools to solve this problem, hence arriving at a hierarchy of SDPs to bound iterated mean divergences (and thus conditional entropies) in device-independent scenarios.

Brown et al. apply their method to obtain new lower bounds on the key rate achievable in various device-independent QKD and randomness generation protocols. Although it is still an open question whether in the asymptotic limit iterated mean divergences converge to the exact relative entropies, the numerical computations carried out show that, for some ranges of noise, their method greatly improves on prior results. Compared to other approaches which go beyond min-entropy estimation, their algorithm does not require large computational resources to produce sensible lower bounds on the key rate for protocols with many-outcome measurements.

All in all, I find the results of this paper very impressive. In a sense, the authors have automatized the process of security certification in the device-independent arena! I therefore recommend the publication of this manuscript in its current form.

Reviewer 2

The authors introduce a family of quantities which they call "iterated mean Renyi divergences" which they define in terms of the solutions to a family of semidefinite programming problems (SDPs - a

type of optimization problem that can be solved efficiently with known algorithms) and which they show provide a lower bound on the conditional von Neumann entropy, which is related to proving the security of cryptography protocols. When the quantum devices are uncharacterized (the so-called device independent scenario) bounding the iterated mean Renyi divergences is not a SDP anymore but it can be relaxed to a hierarchy of increasingly accurate SDPs using a known method (the NPA hierarchy) which can then be solved. The authors then apply their method to obtain key rates and randomness rates for device independent quantum key generation and randomness protocols.

Some context: Device independent cryptography is a type of quantum cryptography in which security is proved based only on the violation of a Bell inequality detected in the protocol. If realized, it would be the most secure type of quantum cryptography possible, since it does not rely on any detailed modeling of the quantum devices which always has the risk of turning out to be insufficiently detailed in a way that could theoretically be hacked by an eavesdropper. However, device independent cryptography (especially DIQKD) is experimentally very difficult to realize (doing a loophole-free Bell experiment is only a precondition). The Entropy Accumulation Theorem published a few years ago showed how one can prove the unconditional security of a device independent protocol assuming one has a lower bound on the conditional von Neumann entropy in a single round. Following this result, there has been some significant interest in finding better ways to lower bound the von Neumann entropy in device independent protocols, the idea being that being able to obtain better key rates or randomness generation rates could lessen the demands on experiments and make a realization of a device independent protocol easier.

The present paper is certainly an interesting contribution in this line and should be published, but I am undecided about whether to recommend this paper for publication in Nature Communications.

The main advantage of the method proposed by the authors is its generality and (relative) simplicity. It is one of only a few methods that have been proposed for numerically bounding the entropy in device independent cryptography that does not depend on the Jordan lemma, a tool used in many cryptography papers that allows the security proof to be reduced to considering qubit systems, but which can only be applied in protocols (such as the one based on CHSH) where the parties only do two measurements with two outputs to test the security. The method proposed by the authors could in principle be applied to protocols with more measurements and/or outputs, and the authors show that they are able to obtain results for protocols where the measurements have three outputs. Compared with existing such numerical approaches, the method outperforms that of bounding the min-entropy using the NPA hierarchy and sometimes, but not always, outperforms another method published on the Arxiv more than a year ago (Ref. [49] in the paper). The method proposed in [49] however is much more computationally intensive than the method that Brown et. al. propose and may not be practical to use except in very small scenarios with few measurements and outputs. However, the practical interest of larger scenarios is unclear to me. Additionally, results for the key rate or randomness generation rate obtained with their method, in section 4, are in some cases better than results obtained using existing methods, however sometimes the reverse is true.

I have no significant concerns about the technical accuracy of the work. I have checked some of what seemed to be the most important proofs and my impression is that the authors know what they are doing.

I only have a few comments on the content of the paper:

- 1) I would have expected the authors to report the result they can obtain for QKD for white noise in a paper like this, or the result for $H(A|E)$ if only the CHSH violation is used as a constraint, since a tight bound for $H(A|E)$ is known for this and it would be interesting to know how close the authors are able to get to it with their method.

Response

We agree with the referee that it is useful to compare directly with the known tight analytical bound on $H(A|E)$. We have added a plot to the Supplementary Information which looks at the local randomness generated with our method when the devices are constrained only by a CHSH score. We compare our bounds to known tight lower bounds on $H(A|E)$ and $H_{\min}(A|E)$. Unfortunately we find in this setting that our method provides negligible improvement over H_{\min} . We believe that this is related to the fact that in this setting there are other entropies over which the infimum coincides with $\inf H_{\min}(A|E)$. For example it has been shown before that the optimization of $\tilde{H}_2^\downarrow(A|E)$ coincides with H_{\min} and we have added a lemma to the Supplementary Information to show that $\tilde{H}_2^\uparrow(A|E)$

also coincides. As $H_{(2)}^\uparrow(A|E)$ is almost the same as $\overline{H}_2^\uparrow(A|E)$ (it differs by the operator inequality $V + V^* \geq 0$) we could well expect that we observe similar behaviour.

2) Why are Alice and Bob's Hilbert spaces labeled " Q_A " and " Q_B " instead of " A " and " B "? I lost some time trying to understand if there was a reason for this and didn't find one.

Response

This notation was used when we had a system that underwent a measurement. We used Q_A to denote the quantum system before the measurement and A to denote the classical system that records the measurement outcomes. We have added a sentence at the end of the preliminaries to explain this notation.

3) I think " I_B " on line 302 should be " I_E ".

Response

Corrected, thanks.